# communications
# earth & environment

# Steam caps in geothermal reservoirs can be monitored using seismic noise interferometry

Pilar Sánchez-Pastor [1,2 ✉], Sin-Mei Wu[1,3], Ketil Hokstad[4], Bjarni Kristjánsson[5], Vincent Drouin [6], Cécile Ducrocq[7], Gunnar Gunnarsson[5], Antonio Rinaldi [1], Stefan Wiemer [1] & Anne Obermann [1]

Harvesting geothermal energy often leads to a pressure drop in reservoirs, decreasing their profitability and promoting the formation of steam caps. While steam caps are valuable energy resources, they also alter the reservoir thermodynamics. Accurately measuring the steam fraction in reservoirs is essential for both operational and economic perspectives. However, steam content estimations are very limited both in space and time since current methods rely on direct measurements within production wells. Besides, these estimations normally present large uncertainties. Here, we present a pioneering method for indirectly sampling the steam content in the subsurface using the ever-present seismic background noise. We observe a consistent annual velocity drop in the Hengill geothermal field (Iceland) and establish a correlation between the velocity drop and steam buildup using in-situ borehole data. This application opens new avenues to track the evolution of any gas reservoir in the crust with a surface-based and cost-effective method.

[1] Swiss Seismological Service (SED), ETH Zürich, Zurich, Switzerland. [2] Geosciences Barcelona (GEO3BCN), CSIC, Lluís Solé i Sabarís s/n, Barcelona, Spain. [3] Earth and Environmental Sciences Area, Lawrence Berkeley National Laboratory, Berkeley, CA, USA. [4] Equinor Research Centre, Arkitekt Ebbells Vei 10, N-7053 Trondheim, Norway. [5] OR—Reykjavik Energy, Bæjarhálsi 1, 110 Reykjavík, Iceland. [6] Icelandic Meteorological Office, 101 Reykjavík, Iceland. [7] Nordic Volcanological Center, Institute of Earth Sciences, University of Iceland, 101 Reykjavík, Iceland. ✉email: psanchezsp@gmail.com

Geothermal energy is often considered renewable and sustainable, although both terms are controversial in the geoscience community. While renewability is related to the natural replenishment of the energy resource, sustainability describes how the resource is used and it involves social, economic, and environmental aspects[1]. The Earth's interior heat and groundwater supplies can be considered renewable. However, in most geothermal fields, the massive fluid extraction is not timely replaced by natural recharge, causing land subsidence[2–4], and a decrease in well productivity and profitability[5], even in the case of partial re-injection of the extracted fluids. Furthermore, geothermal development can lead to chemical pollution, induced seismicity and other environmental impacts[6]. For these reasons, geothermal energy can be considered sustainable when the resources are ensured for the desired life span of the power plant, and their utilization does not compromise future generations.

Most high-enthalpy geothermal reservoirs contain a vapor-rich boiling zone commonly referred to as steam cap. This zone is typically found below a clay-dominated layer, known as cap rock, which effectively traps heat, fluids, and gases in the subsurface due to its low permeability. The steam cap can form through natural cooling of the heat sources or decompression boiling resulting from the massive fluid extraction[7]. An accumulation of steam entails risks of reservoir roof collapse and flooding by colder adjacent fluids[8]. It also strongly influences heat and mass transfer[9], gas transport[10], subsidence[11], and volcano stability[12], among other processes. Furthermore, steam accumulations, especially in wet conditions, represent attractive and profitable resources for electricity generation. Therefore, estimating the steam content within the crust is valuable for both operational and economic perspectives.

In some geothermal systems, gas and liquid phases coexist (two-phase fluids), making it challenging to differentiate the relative fraction of each phase, even from in situ measurements within wells. This challenge arises from the different flow velocities of the gas and liquid phases and from steam flashing along the wellbore[13]. Typically, the steam fraction can be inferred from pressure and temperature measurements taken in monitoring or production wells[14]. Monitoring wells are usually situated in the reservoir periphery, while production wells are directly connected to reservoirs. Prior to measurements in production wells, ~2 weeks of waiting time are required to allow the reservoir reach steady-state conditions. The extended pause in production renders such measurements scarce and costly.

Shallow gas accumulations (<1 km) can be imaged by electromagnetic and gravity surveys[15], although both methods have limitations when it comes to imaging relatively small and deep gas reservoirs[16]. A limited number of active seismic surveys have been conducted to locate gas in the crust[17,18]. However, this approach relies on large number of sensors and extensive field campaigns, making it challenging for long-term continuous monitoring with high temporal resolution. Seismic noise interferometry (SNI) is a surface-based and non-invasive technique that uses recordings of the ever-present seismic noise to extract the seismic response of the Earth's interior[19–21]. This technique enables the study of mechanical and structural variations in the crust with high temporal and spatial resolution[22,23]. These variations are typically quantified through changes in seismic velocity ($\triangle v/v$)[24,25] and waveform similarity[26,27]. SNI has been widely employed to study different processes such as earthquakes[28], volcanic eruptions[29], glaciers[30], and groundwater variations[31], among others. However, establishing a quantitative correlation between surface observations and subsurface steam content remains an unaddressed challenge.

In this study, we present a pilot application to directly monitor steam content in the crust using SNI, enhancing this method beyond its current capabilities. We employ a single-station approach, computing vertical component auto-correlations (ACs), to track the evolution of steam formation in the Hengill geothermal field, Iceland (Fig. 1a). This geothermal field hosts two large power plants: Nesjavellir and Hellisheiði, with the latter ranking as the fifth-largest power plant globally. During the period spanning 2018 and 2021, a total of 55 seismic stations were running in this region as part of the COSEISMIQ project[32] (http://www.coseismiq.ethz.ch). Within this timeframe, we derive time series of $\triangle v/v$ using SNI and surface displacement rates through Interferometric Synthetic Aperture Radar (InSAR) measurements. We first investigate the spatial relationship between surface displacement and mass balance in the reservoir resulting from geothermal operations. Subsequently, we quantitatively assess the thermodynamic evolution of the subsurface based on decadal in situ borehole data. Finally, we predict the evolution of P- ($V_P$) and S-wave velocities ($V_S$) through a rock-physics model and compare it with our surface-based observations.

## Results and discussion

**Impact of geothermal energy production in the crust.** The Hengill geothermal field is located in the easternmost volcanic complex of the Reykjanes Peninsula in Iceland. This area lies at a tectonic triple junction where the North America and the Eurasian tectonics plates diverge (Fig. 1a). In addition, a series of N30°E striking fissure swarms crosses the region, with the Hengill fissure swarm being particularly prominent. These fissure swarms result from the substantial volcanic, tectonic, and seismic activity in the area[33,34]. This unique geological setting offers a wealth of geothermal resources, which are currently targeted by over a hundred deep boreholes (>1 km). In addition to the Nesjavellir and Hellisheiði geothermal fields, there are smaller geothermal fields such as Bitra, Hveragerði, and Selfoss in the vicinity (Fig. 1a). An overview of the existing boreholes in this region and in Iceland is available at https://map.is/os/.

We measure the ground displacement rates in the Hengill area (Fig. 1a) using Sentinel-1 SAR images from 2018 to 2021 (see Methods subsection "Surface deformation"). The results reveal three main centers of subsidence, with maximum values of −1.8, −2 and −1.2 cm/year, located in proximity to the Nesjavellir and Hellisheiði power plants, as well as the Hverahlið geothermal field, respectively. During the same time period, the average mass balance (difference between extracted and re-injected fluid volumes) within the Hellisheiði geothermal field shows a large footprint of fluid extraction in contrast to injection (Fig. 1b). Approximately 60% of the extracted mass is re-injected back into the reservoir[11]. Consequently, and due to the presence of conductive faults[35], the mass deficit and subsidence extend to neighboring areas, affecting regions even with active re-injection. In areas far removed from the production fields, the deformation rates oscillate around zero (Fig. 1a).

We select three seismic stations in the Hellisheiði production field, strategically located in areas with varying extraction rates, and aligned nearly parallel to the fissure swarm (numbered triangles in Fig. 1b). In these locations, we extract decadal temperature and pressure estimates from a hydrogeological model of the Hengill area[36]. This model is based on in situ borehole measurements and it is obtained using the iTOUGH2 software suite[37]. The largest thermodynamic changes occur at ~0.4 km below sea level (~0.8 km below surface) (Fig. 2a), just beneath the nearly impermeable cap rock[38]. At location #1, both pressure and temperature drop by up to 2 MPa and 35 °C, respectively, over a span of nearly 10 years. This results in decompression boiling within the

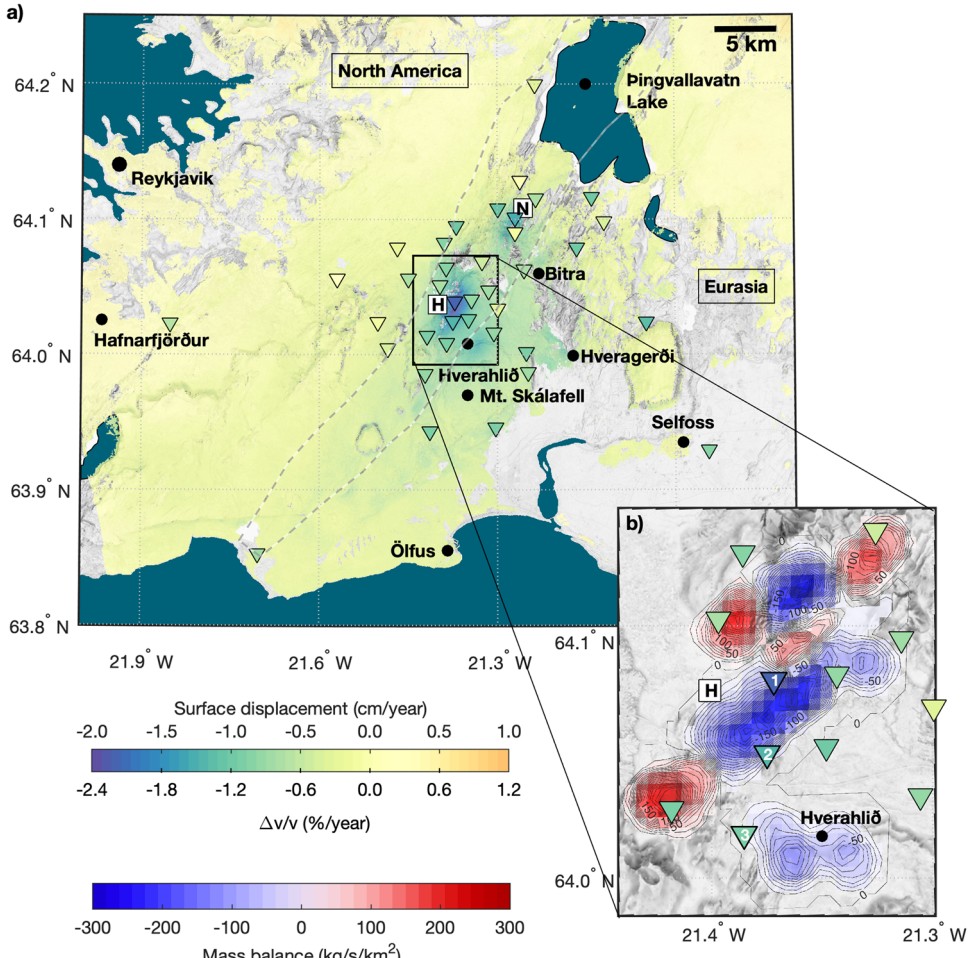

**Fig. 1 Surface observations in the Hengill area. a** Near-up surface displacement rates in the Hengill area. The location of the Nesjavellir and Hellisheiði power plants is represented with their corresponding initials within white squares. The location of the employed seismic stations is depicted by inverted triangles whose colors indicate the annual $\triangle v/v$ rate. The Hengill fissure swarm is delineated with a dashed gray line[63] and places of interest are depicted with solid black dots. **b** Mass balance in the Hellisheiði geothermal field, bluish colors (negative values) indicate production areas and reddish colors (positive values) injection fields. The numbered seismic stations are used in the following figures. The topographic map texture is based on the Digital Elevation Model (DEM) of the area (http://data.europa.eu/88u/dataset/e6712430-a63c-4ae5-9158-c89d16da6361).

reservoir and the formation of a steam cap (Fig. 2b). The steam fraction increases up to 35% at location #1 over the same time period. In contrast, the changes in temperature, pressure, and the subsequent increase in steam are less pronounced at locations #2 and #3, consistent with the lower extracted fluid rates (Fig. 1b).

Subsidence serves an indicator of mass deficit within geothermal systems[2–4]. However, the extent of subsidence hinges on various factors such as the pressure and temperature drawdown, the pore-fluid compressibility, and the elastic rock properties[11]. Despite the largest thermodynamic changes occur below the clay cap, this layer is typically the responsible for most of the subsidence due to its high compressibility[39]. In the case of Hengill, the subsidence is mainly attributed to the pressure drawdown in the reservoir, with thermal rock compaction and plate-boundary spreading playing a lesser role[2]. We suspect that the subsidence rate in Hengill is also influenced by the variations in the steam content, as the compressibility of the rock matrix changes over time. Only in a very few geothermal areas, land subsidence has been linearly correlated to the pressure decay in the reservoir[40,41]. Thus, monitoring the thermodynamic evolution of reservoirs through geodetic methods remains a challenging endeavor.

**Effect of steam on seismic velocities**. To quantify the potential effect of steam on the seismic velocity for P-waves ($V_P$) and S-waves ($V_S$), we construct a rock physics model based on the hydrogeological model of the area[36] (Supplementary Fig. 1a–c). We estimate the bulk and shear moduli depending on temperature[42,43], porosity, and clay fraction[44] assuming hydrostatic conditions. We further include the gas saturation in the model using Gassmann's equations[45,46]. For further details, we refer readers to the "Methods" section, subsection "Rock physics model". The obtained seismic velocity models show a prominent anomaly within the steam cap, where $V_P$ decreases over time and $V_S$ increases with a smaller amplitude. As expected, the larger the extracted fluid mass, the larger the velocity anomalies (Supplementary Fig. 1d, e). The same observations can be made in Fig. 2c, which shows the velocity anomaly at the depth of the largest steam ratio (~0.7 km b.s.l.).

Gassmann's model assumes the shear-wave modulus fluid-independent[46]. For this reason, $V_S$ exclusively depends on density, which decreases 20 times less than the bulk modulus over time (Fig. 3a, b). This makes $V_P$ highly sensitive to fluid phase changes and explains the amplitude difference in the velocity anomaly between both wave types (Fig. 2c and Supplementary Fig. 1d, e). In this way, pressure and temperature

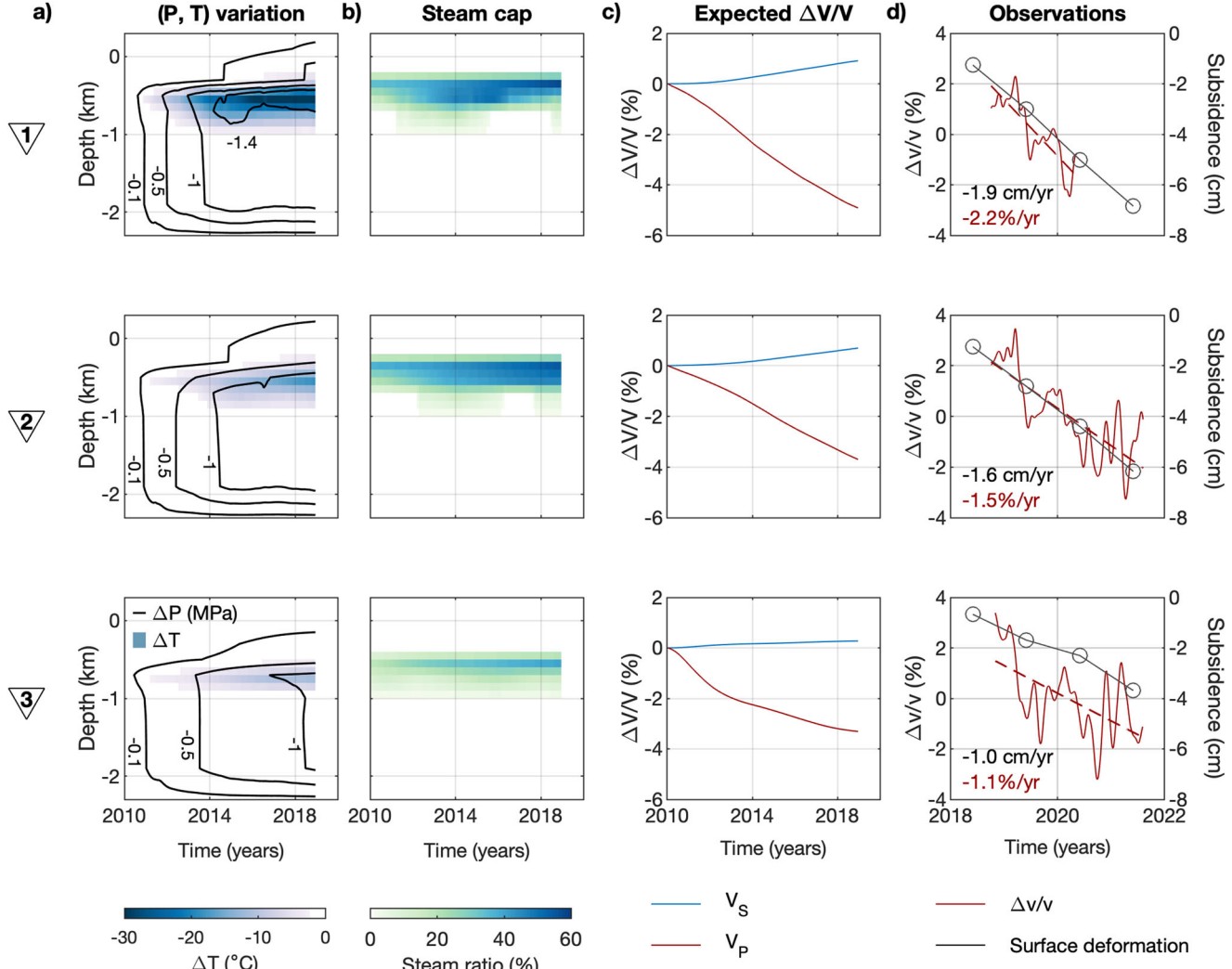

**Fig. 2 Reservoir thermodynamics and surface observations. a** Decadal pressure and temperature variations from the iTOUGH2 hydrogeological model of the area[36] at the three locations numbered in Fig. 1b. **b** Same for the steam ratio. **c** Modeled seismic velocity variations for P- and S-waves obtained from the rock physics model (see Methods subsection "Rock physics model"). **d** Cumulative near-up surface displacement during the COSEISMIQ project (black line) and the observed $\triangle v/v$ (dark red line) together with their annual rate.

variations (included in the density estimations) can be better studied through $V_S$ and phase changes through $V_P$. It is important to mention that rock porosity also controls the effective rock properties and, consequently, the seismic velocities, where a more porous medium undergoes larger changes in elastic properties (Fig. 3).

To validate the model, we calculate the $\triangle v/v$ from the early coda of vertical-component ACs obtained from noise recordings between 2018 and 2021. The data is band-pass filtered between 0.1 and 1 Hz, where seismic waves are sensitivity to reservoir depths (see Methods subsection "Passive seismic monitoring" and Supplementary Fig. 2). Results from the previous three study locations show short- and long-term variations, including a clear decrease over time. Assuming a linear trend, the $\triangle v/v$ exhibits an annual decay of −2.2, −1.5 and −1.1%/year at those locations, respectively (Fig. 2d). The rest of the stations also show a velocity decay (Supplementary Fig. 3), which is overall well correlated with the average subsidence in the area (Fig. 1a). Stations distant from geothermal harnessed areas exhibit a $\triangle v/v$ trend near zero, where subsidence is practically absent.

Both land subsidence and seismic velocity decay are byproducts resulting from the harnessing of geothermal energy in the Hengill region. Consequently, both observables exhibit a similar time evolution (Figs. 1a and 2d). This interrelation underscores a clear correlation between thermodynamic evolution in the reservoir, subsidence rate, and seismic velocities (Fig. 2). We suspect most of the subsidence is caused by rock compaction within the clay cap, as documented in various other geothermal fields[39]. However, geodetic studies have inferred rock compaction within the reservoir itself in Hengill[11], which can induce a marginal reduction in rock porosity, thereby mitigating the seismic velocity decay to some extent (Fig. 3c).

The long-term trends of the observed $\triangle v/v$ closely resembles the modeled $V_P$ evolution (Fig. 2c, d), indicating a high sensitivity to $V_P$ in the early coda of ACs. This finding aligns with the results of a recent study on groundwater systems[47], in which the $\triangle v/v$ series analyzed in ACs are consistent with the outcomes derived from receiver functions. The employed lapse-time and frequency band define the depth sensitivity of the results as well as the energy equipartition ratio[48]. In a place like Hengill, where the subsurface undergoes abrupt variations with depth[49,50] and time (Fig. 2), studying the $\triangle v/v$ dependency with the lapse-time and frequency might not sufficient to distinguish the bearing wave modes and subsurface changes. On the other hand, and despite

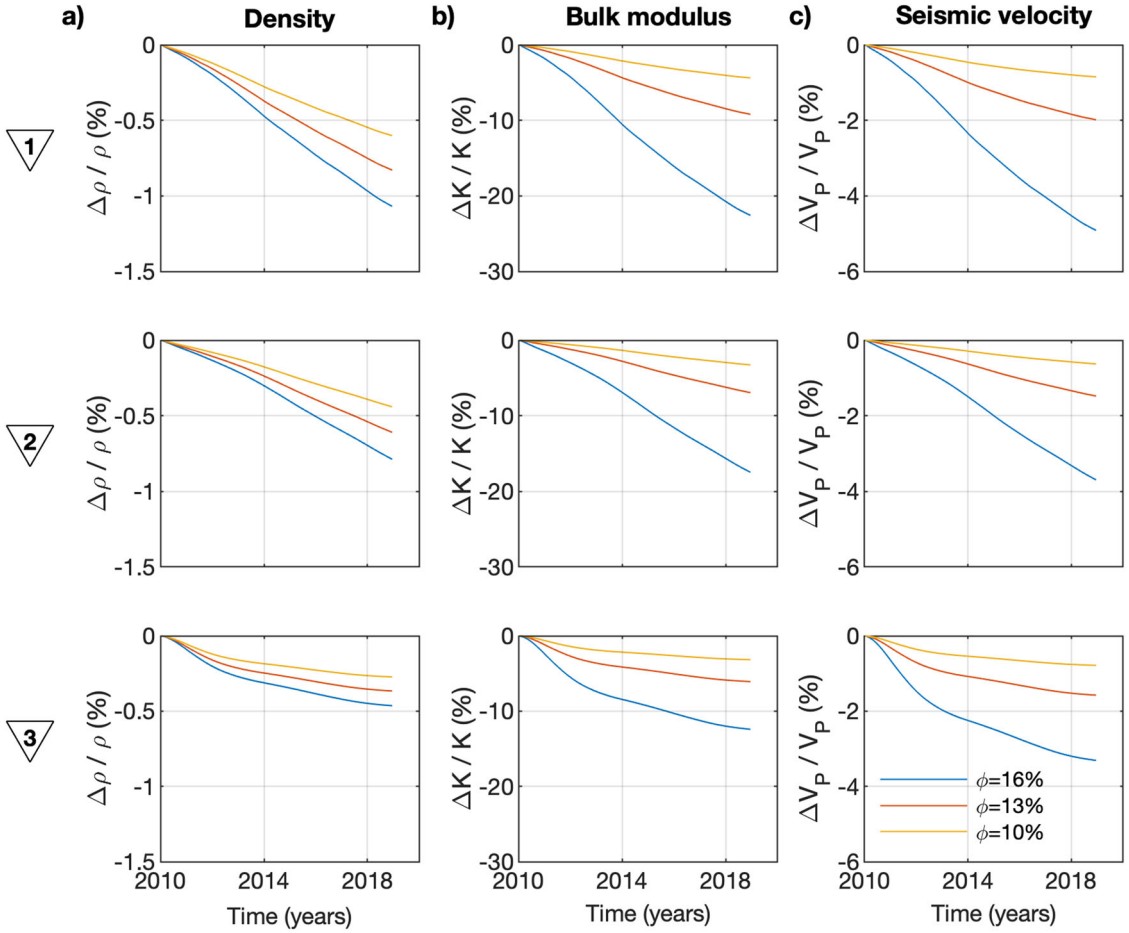

**Fig. 3 Rock physics model.** Modeled (**a**) density, (**b**) bulk modulus, and (**c**) $\triangle V_P/V_P$ for different porosity ($\phi$) values at the three locations numbered in Fig. 1b.

Rayleigh waves are likely present in ACs' coda, they are expected to suffer a slight velocity increase, similar to $V_S$[51]. Therefore, we can conclude that the variations in $V_P$ are the dominant contributors to our $\triangle v/v$ observations.

For these reasons, when studying temperature and pressure variations in reservoirs, we recommend analyzing cross-component ACs and their polarization in order to identify S-waves, which are hardly sensitive to fluid-phase shifts. When studying these instead, ACs of the vertical component of seismic noise proves sufficient due to their remarkable sensitivity to P-waves, which, in turn, are highly responsive to fluid-phase variations.

**Implications.** The harnessing of geothermal energy has a thermodynamic impact on reservoirs, characterized by a decrease in pressure and temperature[5]. These changes can lead to the formation of a steam cap, as observed in Hellisheiði (Fig. 2a, b), as well as rock compaction and subsequent subsidence (Figs. 1a and 2d). The interplay of these thermodynamic and mechanical processes makes reservoir monitoring from indirect and surface measurements, such as surface displacement rates, highly complex. Nevertheless, our results demonstrate the feasibility of monitoring reservoirs from the surface and distinguishing between temperature/pressure and fluid-phase changes by analyzing different wave modes (Fig. 4). Moreover, the single station approach is computationally cost-efficient and can provide near real-time estimations, offering valuable information for optimizing energy production in a timely manner.

Estimating the current steam fraction in the Earth's crust can be achieved in two ways: (1) by using a precise hydrogeological model to fit observed and modeled $\triangle v/v$ over the same time period, and (2) by applying Gassmann's relations[46] with good initial and current $V_P$ estimates. In both cases, an estimation of the porosity and initial steam fraction is required. The observed differences in annual $\triangle v/v$ rates between the model and observations can be attributed to their different estimation periods as well as their distinct initial conditions. Moreover, longer seismic noise recordings can also improve annual rate estimations by averaging out seasonal variations in the observed $\triangle v/v$ time series. A limitation more challenging to tackle is the large uncertainties associated with steam fraction estimations along wellbores[13]. Precise estimations are essential for fine tuning the seismic velocity model and accurately quantifying the current steam fraction within the reservoir. Nevertheless, the results could serve as a foundation for future studies aiming to establish a direct correlation between the $\triangle v/v$ endeavor and the gas fraction in the subsurface.

A similar seismic velocity drop has also been reported in the Reykjanes geothermal system[52]. The physical cause of this velocity decay was attributed to the fluid deficit in the reservoir; however, based on the findings presented in this study, it is possible that the existing steam mass[53] is growing. This phenomenon is observed in numerous geothermal systems[7], where we anticipate $V_P$ to decrease as well. The substantial seismic velocity decay within a matter of years also carries important implications for monitoring induced seismicity within geothermal fields. Therefore, a thorough assessment of seismic

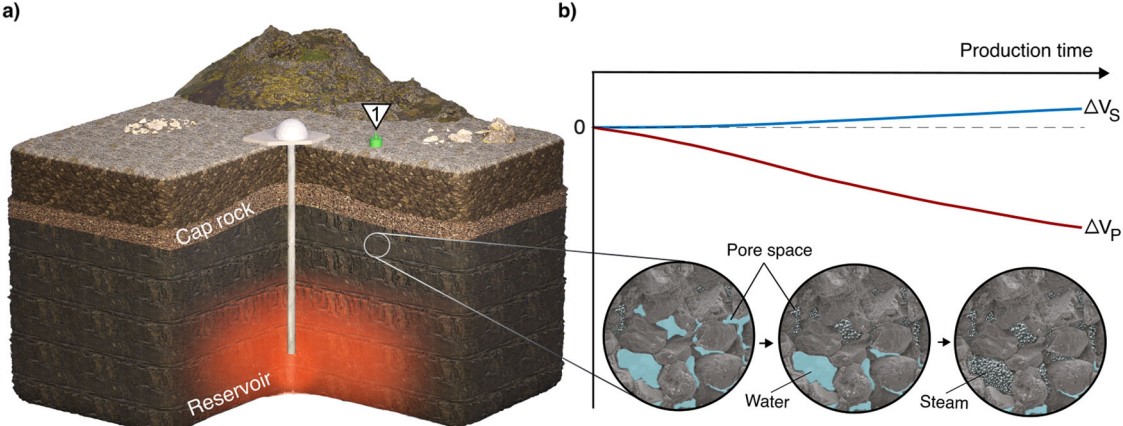

**Fig. 4 Conceptual model of a harnessed geothermal reservoir. a** 3D representation of the geothermal reservoir in the Hellisheiði geothermal field.
**b** Evolution of the pore fluids beneath the cap rock and expected seismic velocity variations observed with one single seismic sensor deployed on the surface.

hazards would require regular updates to seismic velocity models used in seismic location algorithms.

In conclusion, SNI is a surface-based, non-intrusive, and cost-effective method that is well-suited for indirectly sampling gas reservoirs in the Earth's crust using a single seismic sensor (Fig. 4). Furthermore, this approach can be applied to a variety of geological settings. Our findings open up new avenues for investigating the transient behavior of confined gases in the crust and enhancing our understanding of the intricate fluid-rock interactions occurring at depth. Future work will aim to quantify the current steam fraction in the subsurface following the above-mentioned ways.

## Methods

**Surface deformation**. We analyze the SAR images captured by the Sentinel-1 satellites in the Hengill region from 2018 to 2021. To avoid signal loss caused by snow cover during winter, we focus on acquisitions during the summer months, spanning from early June to late September. Deformation time series are generated using the InSAR Scientific Computing Environment (ISCE v2.5.1) and the SBAS code for the Small Baselines Subset Algorithm[54]. The acquisitions include data from ascending orbit 16 and from descending orbit 155, enabling to decompose the signal into near-East and near-Up displacements[55]. The obtained near-Up velocity map is depicted in Fig. 1a.

**Passive seismic monitoring**. We use continuous records from the seismic stations within the COSEISMIQ project (http://www.coseismiq.ethz.ch) to estimate the $\triangle v/v$ time series in the Hengill region. We select a total of 38 seismic stations that were running for more than 1.5 years from October 2018 to August 2021. The workflow followed in this study is detailed below.

- Pre-processing: for each station, we first remove the mean and trend from the continuous seismic records of the vertical component. In cases where there are gaps in the records, we fill those gaps with zeros and merge the different segments. Subsequently, we decimate the data to 10 Hz and convert it into velocity by deconvolving the instrument response across a wide frequency range with corner frequencies (0.001, 0.005, 3, 5) Hz. Finally, we segment the data into 1-h intervals and apply a non-causal band-pass filter of (0.1–1) Hz. Traces with amplitudes larger than $10^{-3}$ m/s are discarded. The entire process is

performed with Python and the ObsPy seismological toolkit[56,57].

- Correlations: we compute the ACs of the filtered 1-h segments using the Phase Cross-Correlation (PCC) method[58] for lag times ranging from −50 to 50 s. The computation of the PCC can be accelerated by rewriting the PCC as a complex cross-correlation[59]. The corresponding code is available at https://github.com/sergiventosa/FastPCC.

- Stacking: we average the hourly correlations by employing a 3-days-length moving window with a 2-days overlap. The use of the PCC enables a quick stabilization of the correlations, a feature well-documented in numerous studies[26,52].

- Seismic velocity estimations: we employ a modified version of the moving-window cross-spectrum (MWCS) analysis[60,61]. With this approach, the velocity changes at each day ($\triangle v_i$) are estimated solving the equation $\triangle v_{ij} = \triangle v_j - \triangle v_i$ by a Bayesian least-squares inversion. $\triangle v_{ij}$ is computed using the MWCS technique for each combination of days $i$ and $j$. This approach ensures that the seismic velocity changes do not depend on any arbitrary reference. However, it is important to note that shortening or enlarging the seismic dataset can affect the velocity-changes time series since the velocity estimations are relative to the dataset used. We apply this technique within the early coda, using a moving lag-time window of 10-s length with an 2/3 overlap, sliding from 10 to 50 s. The estimated $\triangle v/v$ errors[60] are consistently less than 0.1% for all stations, remaining smaller than the background fluctuations in the results.
The method allows selecting a correlation length that defines the temporal distance in the time series where estimations are statistically meaningful. For more in-depth information, we refer readers to the Supplementary Material in the original paper[60]. In this study, we solve the inverse problem using a correlation length of 5 days and subsequently smooth the results by averaging them within a Gaussian-weighted moving window of 1-month length.

- Depth sensitivity: typically, the depth sensitivity of coda waves is estimated through the 1-D sensitivity kernels of Rayleigh waves[62], assuming the dominance of this wave mode in the early correlation coda[48]. While the wave-mode ratio in cross-correlations of ambient seismic noise has been relatively well studied in the past[48], the bibliography for auto-correlations is rather scarce. Our results and a recent study on groundwater systems[47] show the strong influence of P-waves in the $\triangle v/v$ results of vertical-component ACs. Under these circumstances, the presence of body waves provides sensitivity at greater depths[48], and therefore, the depth sensitivity of Rayleigh waves can be used as an estimation of the uppermost layer to which the ACs are sensitive. Using 1D seismic velocity models of the area[50] and for the frequency band (0.1–1) Hz, the observed $\triangle v/v$ time series are sensitive to depths greater than 500 m below surface (Supplementary Fig. 2). This means that, in the Hellisheiði field, the time-series results are sensitive to depths below the clay clap and therefore, to the deep geothermal reservoir. We do not expect the results to be sensitive to more than a few kilometers of depth since the attenuation in this area is very high. Nonetheless, large seismic anomalies are only expected in the top part of the reservoir (Supplementary Fig. 1).

**Rock physics model**. The temperature dependence of the basalt matrix properties is mainly due to volume expansion[42,43]. The relationship between expansivity ($\alpha$) and temperature ($T$) can be expressed as:

$$\alpha(T) = a_0 \left(1 - \frac{10}{\sqrt{T}}\right), \tag{1}$$

which gives

$$\Phi \equiv \ln(V(T)/V_0) = \int_{T_0}^{T} \alpha(T')dT' = a_0 \left[(T - T_0) - 20\left(\sqrt{T} - \sqrt{T_0}\right)\right] \tag{2}$$

where $V(T)$ and $V_0$ are the molar volumes at temperature $T$ and STP respectively and $T_0 = 298K$. Then, the temperature-dependent basalt matrix density $\rho(T)$ can be written as:

$$\rho_m(T) = \rho_0 e^{-\Phi} \tag{3}$$

where $\rho_0$ is the density at temperature $T_0$. The temperature dependence of the matrix bulk and shear moduli $K_m(T)$ and $\mu_m(T)$ are correspondingly given by:

$$K_m(T) = K_0 e^{-\delta_T \Phi}, \tag{4}$$

$$\mu_m(T) = \mu_0 e^{-\Gamma_T \Phi}, \tag{5}$$

where $K_0$ and $\mu_0$ are the bulk and shear moduli at $T_0$, and $\delta_T$ and $\Gamma_T$ are the Gruneisen parameters. The temperature-dependent seismic P- and S-wave velocities of the basalt matrix are given by:

$$V_{P_m}(T) = \sqrt{\frac{K_m(T) + (4/3)\mu_m(T)}{\rho_m(T)}} \tag{6}$$

$$V_{S_m}(T) = \sqrt{\frac{\mu_m(T)}{\rho_m(T)}} \tag{7}$$

To account for the effects of porosity $\phi$, clay fraction $V_{cl}$, and effective stress $\sigma\prime$ on the seismic velocities, we use the following linearized model[44]:

$$V_P(T, \phi, V_{cl}, \sigma') = V_{P_m}(T) - A_P\phi - B_P V_{cl} + C_P(\sigma' - D_P e^{-\beta\sigma'}), \tag{8}$$

$$V_S(T, \phi, V_{cl}, \sigma') = V_{S_m}(T) - A_S\phi - B_S V_{cl} + C_S(\sigma' - D_S e^{-\beta\sigma'}), \tag{9}$$

where $A_\eta, B_\eta, C_\eta, D_\eta$ are empirical coefficients (for $\eta = \{P, S\}$)[44] and the clay fraction is assumed null in this study ($B_\eta = 0$). The effective stress is the lithostatic stress carried by the rock matrix minus the pressure of the pore fluids $P$:

$$\sigma' = \sigma - P \tag{10}$$

In the hydrostatic case, the effective stress is given by:

$$\sigma'(z) = \int_0^z g_z \left[\rho(z', T, \phi, V_{cl}) - \rho_w(T)\right] dz' \tag{11}$$

where $g_z$ is the acceleration of gravity, and $\rho_w$ is the temperature-dependent water density.

The bulk density is then given by:

$$\rho(T, \phi, V_{cl}) = (1 - \phi - V_{cl})\rho_m(T) + V_{cl}\rho_{cl} + \phi\rho_w(T), \tag{12}$$

where $\rho_{cl}$ is the clay density.

Assuming that the equations above represent the case of fractures and pores filled with liquid water, we perform fluid substitution with the Gassmann equations[45,46] to model the case when liquid water is partly replaced by steam, with steam fraction ($S_g$) $0 \le S_g \le 1$. The bulk modulus for $S_g$ can be obtained as:

$$\hat{K}\left(T, \phi, \sigma', S_g\right) = \frac{h}{1 + h}K_m \tag{13}$$

where

$$h = \frac{K}{K_m - K} + \frac{1}{\phi}\left[\frac{K_f}{K_m - K_f} - \frac{K_w}{K_m - K_w}\right] \tag{14}$$

and $K = K\left(T, \phi, V_{cl}, \sigma'\right)$ is the bulk modulus of porous basalt with pore space filled with water.

The bulk modulus of the fluid with partial steam saturation is computed as the Reuss average:

$$\frac{1}{K_f} = \frac{S_g}{K_g} + \frac{1 - S_g}{K_w} \tag{15}$$

The shear modulus is independent of the pore fluid:

$$\hat{\mu}\left(T, \phi, V_{cl}, \sigma', S_g\right) = \mu\left(T, \phi, V_{cl}, \sigma'\right) \tag{16}$$

The density becomes:

$$\hat{\rho}\left(T, \phi, V_{cl}, S_g\right) = (1 - \phi - V_{cl})\rho_m(T) + V_{cl}\rho_{cl} + \phi\rho_f(T), \tag{17}$$

where

$$\rho_f(T) = \left(1 - S_g\right)\rho_w(T) + S_g\rho_g(T), \tag{18}$$

and $\rho_g$ is the temperature (and pressure) dependent density of the steam. The seismic P-wave and S-wave velocities are again given by Eqs. (6) and (7).

We use the pressure, temperature and steam fraction models[36] from 2010 to 2018 computed using the iTOUGH2 software suite[37]. The model is calibrated against formation temperature and pressure curves as well as production history data, such as enthalpy of produced fluid and pressure drawdown. The porosity in the model is considered constant with depth and with a value of 10%. We use those models and the input parameters

**Table 1 Input parameters for the rock physics model.**

| Parameter | Value | Units | Comment |
|---|---|---|---|
| $\phi$ | 0.1, 0.13, 0.16 | – | Porosity |
| $\rho_{w_0}$ | 1000 | $kg/m^3$ | Water density at STP[a] |
| $K_w$ | 2 | GPa | Bulk modulus of water |
| $K_g$ | $0.8P_h$ | MPa | Bulk modulus of steam (propoptional to hydrostatic pressure) |
| $\delta_T$ | 2.5 | – | Gruneisen parameter for bulk modulus |
| $\Gamma_T$ | 4 | – | Grunesien parameter for shear modulus |
| $\rho_0$ | 3000 | $kg/m^3$ | Density of basalt at STP |
| $V_{P_0}$ | 6500 | m/s | P-wave velocity of basalt at STP |
| $V_{S_0}$ | 3500 | m/s | S-wave velocity of basalt at STP |

[a]STP stands for standard temperature and pressure conditions, which are 25 °C and 1 atm.

summarized in Table 1 to generate the seismic velocity models (Fig. 2c and Supplementary Fig. 1d, e). The reference velocity used to estimate the relative changes is the one calculated at 2010. Note that the observed velocity changes are computed with no reference period.

## Data availability
The seismic dataset from the COSEISMIQ project is openly available on EIDA (http://eida-federator.ethz.ch/fdsnws/station/1/query?net=2C,OR,VI&format=text&level=station&nodata=404). The seismic networks used in this study are: 2C, OR and VI (http://www.coseismiq.ethz.ch/en/dissemination/stations/). The Sentinel-1 images used to estimate the near-up surface displacements are accessible on the Copernicus and European Space Agency (ESA) platform https://dataspace.copernicus.eu/browser. The Digital Elevation Model (DEM) used in Fig. 1 can be downloaded from https://atlas.lmi.is/mapview/?application=DEM.

## Code availability
The code to reproduce the seismic noise correlations presented in the study is available on https://github.com/sergiventosa/FastPCC.

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

## Acknowledgements

The presented work has been accomplished within the framework of the Geothermica project Derisking exploration for geothermal plays in magmatic environments (DEEPEN; Grant Number 03EE4018). The COSEISMIQ project has been subsidized through the ERANET Cofound GEOTHERMICA (Project no. 731117), from the European Commission, SFOE (SI/501721) and Rannís (170767-4401). P.S.-P. received funding from the European Research Council (ERC) under the European Union's Horizon 2020 research and innovation program Real-Time Earthquake Risk Reduction for a Resilient Europe (RISE; Grant Number 821115) and from the Swiss National Science Foundation (Grant Number 200021-181986). S.-M.W. has been supported by the DEEPEN project. C.D. was supported by the NordVulk fellowship programme (2020) and University of Iceland fund (2021). The authors would like to thank Luigi Pasarelli for the fruitful discussions.

## Author contributions

P.S.-P.: conceptualization, methodology, software, investigation and visualization. S.-M.W.: conceptualization, methodology, software, investigation and visualization. K.H.: methodology, software and investigation. B.K.: methodology and investigation. V.D.: methodology and investigation. C.D.: methodology and investigation. G.G.: methodology and investigation. A.R.: methodology and investigation. S.W.: conceptualization and project supervision. A.O.: oversight. All authors contributed to the writing and review of the manuscript.

## Competing interests

The authors declare no competing interests.
