## [Peer Review File · Communications Earth & Environment]

31st Jul 23

Dear Dr Sánchez-Pastor,

Please allow us to apologise for the delay in sending a decision on your manuscript titled "Monitoring gas pockets in the crust with seismic noise". It has now been seen by 3 reviewers, and we include their comments at the end of this message. They find your work of interest, but some important points are raised. We are interested in the possibility of publishing your study in Communications Earth & Environment, but would like to consider your responses to these concerns and assess a revised manuscript before we make a final decision on publication.

We therefore invite you to revise and resubmit your manuscript, along with a point-by-point response that takes into account the points raised. Please highlight all changes in the manuscript text file.

Please use the following link to submit your revised manuscript, point-by-point response to the referees' comments (which should be in a separate document to any cover letter), a tracked-changes version of the manuscript (as a PDF file) and the completed checklist:

[link redacted]

We hope to receive your revised paper within six weeks; please let us know if you aren't able to submit it within this time so that we can discuss how best to proceed. If we don't hear from you, and the revision process takes significantly longer, we may close your file. In this event, we will still be happy to reconsider your paper at a later date, as long as nothing similar has been accepted for publication at Communications Earth & Environment or published elsewhere in the meantime.

Please do not hesitate to contact us if you have any questions or would like to discuss these revisions further. We look forward to seeing the revised manuscript and thank you for the opportunity to review your work.

Best regards,

Joe Aslin

Senior Editor,
Communications Earth & Environment
<https://www.nature.com/commsenv/>
Twitter: @CommsEarth

EDITORIAL POLICIES AND FORMATTING

Editorial Policy: [Policy requirements](https://www.nature.com/documents/nr-editorial-policy-checklist.pdf) (Download the link to your computer as a PDF.)

Furthermore, please align your manuscript with our format requirements, which are summarized on the following checklist:

[Communications Earth & Environment formatting checklist](https://www.nature.com/documents/commsj-phys-style-formatting-checklist-article.pdf)

and also in our style and formatting guide [Communications Earth & Environment formatting guide](https://www.nature.com/documents/commsj-phys-style-formatting-guide-accept.pdf).

*** DATA: Communications Earth & Environment endorses the principles of the Enabling FAIR data project (<http://www.copdess.org/enabling-fair-data-project/>). We ask authors to make the data that support their conclusions available in permanent, publically accessible data repositories. (Please contact the editor if you are unable to make your data available).

All Communications Earth & Environment manuscripts must include a section titled "Data Availability" at the end of the Methods section or main text (if no Methods). More information on this policy, is available at <http://www.nature.com/authors/policies/data/data-availability-statements-data-citations.pdf>.

If a community resource is unavailable, data can be submitted to generalist repositories such as [figshare](https://figshare.com/) or [Dryad Digital Repository](http://datadryad.org/). Please provide a unique identifier for the data (for example a DOI or a permanent URL) in the data availability statement, if possible. If the repository does not provide identifiers, we encourage authors to supply the search terms that will return the data. For data that have been obtained from publically available sources, please provide a URL and the specific data product name

in the data availability statement. Data with a DOI should be further cited in the methods reference section.

REVIEWER COMMENTS:

Reviewer #1 (Remarks to the Author):

I enjoyed reading this paper which makes the necessary effort to quantify observed velocity changes through thermo-poro-elastic modeling.

I mostly agree with the interpretations and conclusions of the paper, which are well supported by the observations.

There is one interpretation, though, that I feel deserves more in-depth analysis before a possible acceptance of the manuscript: the nature of the ZZ auto-correlation coda and the resemblance between the observed dv/v trends and the modeled dvp/vp trends rather than dvs/vs trends. The authors conclude that the coda should be dominated, even at short lag times, by P-waves rather than Rayleigh waves. Yet there is a factor 2 to 4 between the observed dv/v (-2%/yr) and the modeled dvp/vp (-0.5%/yr). The uncertainties of the data are briefly discussed in the text but never quantified. A similar observation can be made for the modeled dv/v . The supplementary material's last figure clearly shows the porosity's strong influence on the magnitude of the modeled dv/v .

1) It would be really interesting to see how the uncertainties on the model parameter values propagate to the modeled dv/v . A "Monte-Carlo" exploration of the model parameters between "reasonable" bounds will lead to a posterior distribution of dv/v , facilitating the estimation of the modeled trends uncertainties. Perhaps when taking into account the uncertainties on the linear regression of the data dv/v and the uncertainties on the modeled dv/v , the data and model dv/v trends won't be incompatible anymore.

2) Given the fact that you have a relatively robust velocity model of the subsurface (provided that you can also have a 1D density model), I would strongly advise computing synthetic fundamental mode Rayleigh wave phase velocity dispersion curves for the frequency range you are looking at. At the coda lag-time you are investigating, there still must be a substantial contribution of Rayleigh waves, and therefore a comparison with the data dv/v should be made with modeled dispersion curves rather than Vp and Vs separately.

3) About the relative ratio between Rayleigh waves and body-waves in the coda, given the fact that the expected changes are confined in a well-defined layer, there should be a clear frequency and also lag-time dependence of the measured dv/v . Did you check that?

4) In the same objective, looking at the AC of the horizontal components (both EE, NN but also EN), which should also exhibit a significant sensitivity of Love waves, with much less sensitivity from P-wave (virtually none), you should observe a drastically different dv/v if dominated by surface waves.

On the other hand, if the dv/v from the EN component is similar to the ZZ component, then it might be valid to bring the equipartitioned field argument with a relatively equal contribution of all modes of waves.

Minor comments:

5) Line 262: define Phi

6) Line 284: missing g_z

7) Figure 2), Panel c) and d): use the same vertical axis in c) and d) and/or show the value of the modeled dv/v trends for v_p and v_s (and also Rayleigh waves taking into account my comment 2) above.

Reviewer #2 (Remarks to the Author):

The manuscript "Monitoring gas pockets in the crust with seismic noise" by Sánchez-Pastor used ambient noise seismic interferometry to evaluate seismic velocity changes with the aim to estimate current steam at Hengill geothermal field, Iceland. The authors included their analysis from geodetic modeling with InSAR data and also performed numerical simulation with iTOUGH2 software to evaluate possible seismic velocity change (V_p , V_s) at seismic stations that were used for the seismic interferometry analysis. The authors found that seismic velocity changes (dv/v) were gradually decreased for the almost all stations nearby the Hellisheiði power plant, and this result appears to be spatially correlated with the ground displacement from InSAR. I think that this manuscript provides interesting observations and discussion, however I have several questions, which may help to clarify the authors' findings and argument.

My moderate comment is about the interpretation of dv/v (Figure 2). The authors argued that the long-term decrease of dv/v observation is similar to the temporal change of V_p expected from the iTOUGH2 modeling, and therefore, they argued that dv/v of their observation is tracking V_p changes. I feel that there would be more discussion and analysis to argue this. My understanding is that it is a very challenging to extract V_p changes through ambient noise cross correlation, especially the frequency band for ocean microseism (0.1-1.0Hz) that was used in this study. I understand that there would be some parts of body waves (P and S) will be extracted from ambient noise correlation but this may require many stacking to extract them as the most dominant seismic waves would be surface waves. I think that may be at least two approaches to clarify if observed dv/v would be controlled by V_p .

First, the body waves will be more dominantly seen in later parts of noise cross correlation data (Obermann et al, 2013 GJI). I would suggest exploring variations of dv/v with different lapse times to see resultant dv/v from different lapse times will be consistent with V_p changes.

The second part will be to evaluate different frequency bands to see if there is no dispersive characteristics of resultant dv/v . My understanding is that V_p (body wave) change will be less likely to show frequency-dependent changes compared with surface waves.

Please consider these two parts to clarify the V_p change part.

I have to admit that I am not expertise for InSAR and iTOUGH2 modeling parts but when I read this manuscript, these parts were appropriately performed. Also the rock physics section is very useful and it is indeed great to see seismic velocity changes with different temperature and porosity.

Reviewer #3 (Remarks to the Author):

I found this to be an excellent manuscript. The authors definitely have the needed expertise to analyze the data and to draw the given conclusions.

The necessary background, mythology, and implications are clearly presented.

As far as this reviewer knows, the work and methods appear to be novel and appropriate. They use an extensive data set to calibrate and test their methods. As pointed out in the manuscript, calibration is critical because of the many high resolution compositional and spatial variations that rock physics models cannot not completely account for.

I expect that the work will be of interest to a substantial audience and is a valuable contribution to the field.

Reviewer #1 (Remarks to the Author):

I enjoyed reading this paper which makes the necessary effort to quantify observed velocity changes through thermo-poro-elastic modeling.

I mostly agree with the interpretations and conclusions of the paper, which are well supported by the observations.

There is one interpretation, though, that I feel deserves more in-depth analysis before a possible acceptance of the manuscript: the nature of the ZZ auto-correlation coda and the resemblance between the observed dv/v trends and the modeled dvp/vp trends rather than dvs/vs trends. The authors conclude that the coda should be dominated, even at short lag times, by P-waves rather than Rayleigh waves. Yet there is a factor 2 to 4 between the observed dv/v (-2%/yr) and the modeled dvp/vp (-0.5%/yr). The uncertainties of the data are briefly discussed in the text but never quantified. A similar observation can be made for the modeled dv/v . The supplementary material's last figure clearly shows the porosity's strong influence on the magnitude of the modeled dv/v .

1) It would be really interesting to see how the uncertainties on the model parameter values propagate to the modeled dv/v . A "Monte-Carlo" exploration of the model parameters between "reasonable" bounds will lead to a posterior distribution of dv/v , facilitating the estimation of the modeled trends uncertainties. Perhaps when taking into account the uncertainties on the linear regression of the data dv/v and the uncertainties on the modeled dv/v , the data and model dv/v trends won't be incompatible anymore.

We greatly appreciate the positive feedback on our paper and the insightful comments. The suggestions regarding the consideration of uncertainties in our model are indeed valuable.

The seismic velocity models we derive rely on input variables such as pressure, temperature and steam ratio, computed using the iTOUGH2 software suite. This software incorporates the measurements performed in production and monitoring boreholes and generates a continuous 3D hydrogeological model in the area. Besides the uncertainties of the inversion process, the measurements themselves have associated both errors and uncertainties that are difficult to quantify due to the complexity of the measurement procedures within deep geothermal reservoirs. The most important variable for the velocity models is the steam ratio, which is in turn the most complicated to take. Steam ratio measurements are in fact estimations that are inferred from the temperature and pressure measurements (see L69-77). The uncertainties are known to be large but there is no a reliable quantification of them. Additionally, we show in *Extended Data Fig. 2* that the rock porosity has a significant impact on the models as the reviewer certainly points out. However, the rock porosity is evaluated from *in-situ* rock samples that are taken when drilling the boreholes and the uncertainty is also unknown. We do know the plausible range of rock porosity values that are the ones tested in the figure. In this way, analyzing the error propagation into the velocity models is not possible in this study case.

Regarding the dv/v observations, we agree with the concern related to the absence of uncertainty estimations in the manuscript and we have addressed it. Following the approach outlined in the supplementary material of Brenguier et al. (2014), we have now calculated the uncertainties for all seismic stations. The uncertainties consistently fall below 0.1%, smaller magnitude than the background fluctuations of the dv/v results. We included this information in the manuscript (see L848-849). However, it is essential to recognize that dv/v estimations inherently possess uncertainties that are challenging to quantify precisely. When working with coda waves, seismic velocity changes are relative measurements influenced by various factors, including the noise source distribution (e.g., Zhan et al. 2013), the quantification technique and the tune parameters of such technique (e.g., Hadziioannou et al. 2009). As such, providing a comprehensive uncertainty estimation for the observed dv/v is limited by the inherent complexities of coda waves.

Furthermore, obtaining the most recent hydrogeological model for the Hellisheiði geothermal field was not feasible due to the constraints related to data sensitivity. As a consequence, we could not obtain the model and seismic data within the same time frame. Given these limitations and constraints, achieving a fine-tuned fit of the model to the data falls beyond the scope of the current study.

Nonetheless, we would like to emphasize that the idea of incorporating the uncertainties to better align the model and observations is excellent. In the future, we aim at performing laboratory experiments, where we can have better control over the system to fine-tune the model and quantitatively assess the steam ratio. Addressing uncertainties in these controlled settings will indeed be valuable, and we appreciate the reviewer's encouragement in this regard.

2) Given the fact that you have a relatively robust velocity model of the subsurface (provided that you can also have a 1D density model), I would strongly advise computing synthetic fundamental mode Rayleigh wave phase velocity dispersion curves for the frequency range you are looking at. At the coda lag-time you are investigating, there still must be a substantial contribution of Rayleigh waves, and therefore a comparison with the data dv/v should be made with modeled dispersion curves rather than V_p and V_s separately.

While the wave-mode ratio in cross-correlations of ambient seismic noise has been relatively well studied in the past (e.g., Obermann et al. 2013a), the bibliography for auto-correlations is rather scarce. We initially interpreted the resemblance between the velocity model for P waves and the dv/v observations as an indicator of the dominance of P-waves in the coda of our auto-correlations. However, the reviewers' comments led us to re-consider our interpretation and we agree with the fact that we have not performed sufficient analysis to make that assertion.

Nevertheless, the dv/v observations positively correlate with the model for P-waves and anti-correlate with the one for S-waves (Figure 2). We believe the results certainly show the high sensitivity of auto-correlations to P-wave velocities. In fact, this has been proved by a previous study on groundwater systems, where the authors compare the dv/v evolution obtained from seismic noise auto-correlations and receiver functions (Kim and Lekic, 2019). We believe this interpretation is more accurate to our study. Note that the slight interpretation change does not compromise the results and robustness of the study. We have modified the text in the manuscript accordingly (L489-492).

The phase velocity of Rayleigh waves strongly depends on V_s and hardly depends on V_p and density. As we have shown in Figure 2 and Extended Data Fig.2, the variations that V_s and density suffer due to the fluid-phase changes are much smaller than V_p . For this reason and regardless of the wave-mode ratio, the expected contribution of Rayleigh waves to the dv/v evolution is minor. A recent study on seismic signatures of steam in reservoirs (Quiroga et al. 2023) has proved that the group and phase velocities for Rayleigh waves are hardly sensitive to fluid content changes. Given the steam content range estimated in the Hellisheiði, the expected phase velocity change is approximately 0.4% considering a poroelastic approach, as we do in our study. We added relevant information in the manuscript (L496-499).

Nevertheless, we have estimated the Rayleigh wave phase velocity dispersion curves in the frequency band of the study and using the 1D V_p and V_s models in the area (at station's location #1). We add a perturbation of -5% in V_p , +1% in V_s , and -1% in density between 0.5 and 1.5 km depth. The original and the new dispersion curves are represented in red and blue respectively in Figure R1. The velocity perturbation due to the fluid-phase changes causes a very small decrease in the phase velocity of Rayleigh waves, in line with the Quiroga et al. (2023).

Figure R1. The fundamental mode Rayleigh wave dispersion curves (right) calculated with the 1D velocity models (left). The initial and perturbed conditions are shown in red and blue colors, respectively. In the perturbed model, V_p (solid line) is decreased by 5%, V_s (dashed line) is increased by 1%, and density is decreased by 1% at depths of 0.5-1.5 km.

For these reasons, we believe the interpretation of the high sensitivity of ZZ auto-correlations to P-wave velocities is well grounded and despite further analysis on the wave-mode propagation is of interest, it falls outside the scope of this study.

3) About the relative ratio between Rayleigh waves and body-waves in the coda, given the fact that the expected changes are confined in a well-defined layer, there should be a clear frequency and also lag-time dependence of the measured dv/v . Did you check that?

We agree with this comment and it is indeed a good point as well. However, in a complex and dynamic system like Hengill, the thermodynamic state, mineral alteration and lithology abruptly changes with depth (e.g., Sánchez-Pastor et al. 2021). For this reason, studying the different behavior of the results at different frequencies/lag-times will not provide enough insights to distinguish the bearing wave modes and the subsurface changes across depths.

Nevertheless, we did check the frequency and lag-time dependency of the results. As an example, the negative decay in dv/v observations becomes positive at higher frequencies for most of the seismic stations (Figure R2). We expect the seismic waves at frequencies (1 – 2) Hz to be sensitive to approximately the clay cap depths, where we expect most of the rock compaction caused by the pressure drop (see L484-488). The rock compaction would explain the seismic velocity increase, as has been observed in other areas (e.g., Mao et al. 2022).

This result indicates a shallower sensitivity at higher frequency which seems to show a signature of surface waves at those frequencies. When computing dv/v time series, the selected lag-time window is the same for both frequency bands. Given a lag-time window, the frequency band determine the scattering regime of the seismic waves and therefore, the equipartition ratio. Furthermore, Quiroga et al. (2022) show that body waves and Rayleigh waves velocities are highly responsive to fracture density variations, which might be an important factor to consider in the clay cap due to the rock matrix compaction. To discern the contribution of the different wave modes and the subsurface changes at different depths, another specific rock physics models will be needed.

Figure R2. Time series of seismic velocity changes for two different frequency bands (blue line). The results are smoothed with a gaussian window of 2 different lengths and represented by the orange and yellow lines. The black line represents the linear fit of the results.

4) In the same objective, looking at the AC of the horizontal components (both EE, NN but also EN), which should also exhibit a significant sensitivity of Love waves, with much less sensitivity from P-wave (virtually none), you should observe a drastically different dv/v if dominated by surface waves. On the other hand, if the dv/v from the EN component is similar to the ZZ component, then it might be valid to bring the equipartitioned field argument with a relatively equal contribution of all modes of waves.

As Love and Rayleigh waves exhibit distinct depth sensitivities within the same frequency bands, the retrieved dv/v will be nonunique to distinguishing the energy partition and subsurface changes.

We would like to emphasize that we exclusively focus on the thermodynamic changes that occur at the top of the deep reservoir, where the steam cap is located. In this study, we analyze the impact that fluid-phase variations have on seismic velocities and introduce an approach to track it with a surface and non-invasive technique. Therefore, this study does not encompass analyzing other geodynamic processes.

Nonetheless, we value the important point raised and we have added relevant context in the manuscript (L492-496).

Minor comments:

5) Line 262: define Phi **Done**

6) Line 284: missing g_z **Done**

7) Figure 2), Panel c) and d): use the same vertical axis in c) and d) and/or show the value of the modeled dv/v trends for v_p and v_s (and also Rayleigh waves taking into account my comment 2) above.

The iTOUGH2 model and seismic observations do not share the same time frame and we prefer to avoid a direct comparison between the velocity model and observations.

Reviewer #2 (Remarks to the Author):

The manuscript “Monitoring gas pockets in the crust with seismic noise” by Sánchez-Pastor used ambient noise seismic interferometry to evaluate seismic velocity changes with the aim to estimate current steam at Hengill geothermal field, Iceland. The authors included their analysis from geodetic modeling with InSAR data and also performed numerical simulation with iTOUGH2 software to evaluate possible seismic velocity change (V_p , V_s) at seismic stations that were used for the seismic interferometry analysis. The authors found that seismic velocity changes (dv/v) were gradually decreased for the almost all stations nearby the Hellisheiði power plant, and this result appears to be spatially correlated with the ground displacement from InSAR. I think that this manuscript provides interesting observations and discussion, however I have several questions, which may help to clarify the authors’ findings and argument.

My moderate comment is about the interpretation of dv/v (Figure 2). The authors argued that the long-term decrease of dv/v observation is similar to the temporal change of V_p expected from the iTOUGH2 modeling, and therefore, they argued that dv/v of their observation is tracking V_p changes. I feel that there would be more discussion and analysis to argue this. My understanding is that it is a very challenging to extract V_p changes through ambient noise cross correlation, especially the frequency band for ocean microseism (0.1-1.0Hz) that was used in this study. I understand that there would be some parts of body waves (P and S) will be extracted from ambient noise correlation but this may require many stacking to extract them as the most dominant seismic waves would be surface waves. I think that may be at least two approaches to clarify if observed dv/v would be controlled by V_p .

First, the body waves will be more dominantly seen in later parts of noise cross correlation data (Obermann et al, 2013 GJI). I would suggest exploring variations of dv/v with different lapse times to see resultant dv/v from different lapse times will be consistent with V_p changes.

The second part will be to evaluate different frequency bands to see if there is no dispersive characteristics of resultant dv/v . My understanding is that V_p (body wave) change will be less likely to show frequency-dependent changes compared with surface waves.

Please consider these two parts to clarify the V_p change part.

We appreciate the reviewer's comments and suggestions to improve our study. The main concern is related the interpretation of P-wave dominance in the coda, which aligns with the first reviewer's query. After revisiting

the issue, we acknowledge the limited observations that prevent us from definitively asserting P-wave dominance in the coda. However, our findings indeed exhibit a strong resemblance to the P-wave velocity model. This observation can be interpreted as a manifestation of the high sensitivity of ZZ auto-correlations to P-wave velocities, a notion supported by Kim and Lekic (2019).

In this study, the authors analyzed cross-component auto-correlations of seismic noise to study groundwater variations and found that ZZ auto-correlations are highly sensitive to P-wave velocities. Furthermore, Quiroga et al. (2023) has proved that Rayleigh wave phase velocities are less significantly less sensitive to steam content variations than P-waves. As we have discussed in previous responses, whether P-waves dominate the coda or the coda exhibits high sensitivity to P-waves, the overall interpretation of our results remains consistent and robust. We have further clarified this point in the manuscript (see L489-504).

We appreciate the ideas to explore variations in dv/v with different lapse times and evaluate different frequency bands. However, the frequency content and lapse-time window define the depth sensitivity of the dv/v results. In a place like Hengill, where the subsurface undergoes abrupt variations with depth (e.g., Sánchez-Pastor et al., 2021), such analyses may not yield substantial insights into the coda's wave nature without additional modeling for different parts of the subsurface system that subject to different thermo/mechanical processes. This study is driven by having the in-situ data within the reservoir, and nevertheless focuses on the fluid phase changes associated with steam growth in the geothermal system. Please also see our reply to points 2 and 3 commented by reviewer 1. We hope these explanations clarify our approach and address the concerns raised by the reviewer.

I have to admit that I am not expertise for InSAR and iTOUGH2 modeling parts but when I read this manuscript, these parts were appropriately performed. Also the rock physics section is very useful and it is indeed great to see seismic velocity changes with different temperature and porosity.

We appreciate the positive feedback.

Reviewer #3 (Remarks to the Author):

I found this to be an excellent manuscript. The authors definitely have the needed expertise to analyze the data and to draw the given conclusions.

The necessary background, mythology, and implications are clearly presented.

As far as this reviewer knows, the work and methods appear to be novel and appropriate. They use an extensive data set to calibrate and test their methods. As pointed out in the manuscript, calibration is critical because of the many high resolution compositional and spatial variations that rock physics models cannot not completely account for.

I expect that the work will be of interest to a substantial audience and is a valuable contribution to the field.

We appreciate the positive feedback and the assessment of our manuscript. The comments are encouraging, and we are pleased to hear that study is perceived as valuable and novel. We value the time and review.

Bibliography

F. Brenguier, M. Campillo, T. Takeda, Y. Aoki, N. M. Shapiro, X. Briand, K. Emoto and H. Miyake, "Mapping pressurized volcanic fluids from induced crustal seismic velocity drops," *Science*, vol. 345, p. 80–82, 2014.

C. Hadziioannou, E. Larose, O. Coutant, P. Roux & M. Campillo, "Stability of monitoring weak changes in multiply scattering media with ambient noise correlation: Laboratory experiments", *The Journal of the Acoustical Society of America*, 125(6), 3688-3695, 2009

D. Kim and V. Lekic, "Groundwater variations from autocorrelation and receiver functions," *Geophysical Research Letters*, vol. 46, p. 13722–13729, 2019.

S. Mao, A. Lecointre, R. Van Der Hilst and M. Campillo, "Space-time monitoring of groundwater fluctuations with passive seismic interferometry," *Nature Communications*, vol. 13, 2022.

Gabriel E. Quiroga, J. Germán Rubino, Santiago G. Solazzi, Nicolás D. Barbosa, Marco Favino, Klaus Holliger, “Effects of fracture connectivity on Rayleigh wave dispersion.” *Journal of Geophysical Research: Solid Earth*, 127, e2021JB022847, 2022.

Gabriel E. Quiroga, J. Germán Rubino, Santiago G. Solazzi, Nicolás D. Barbosa, Marco Favino, Klaus Holliger, “Seismic signatures of partial steam saturation in fractured geothermal reservoirs: Insights from poroelasticity.” *Geophysics*; 88 (5): WB89–WB104, 2023.

P. Sánchez-Pastor, A. Obermann, T. Reinsch, T. Ágústsdóttir, G. Gunnarsson, S. Tómasdóttir, V. Hjörleifsdóttir, G.P. Hersir, K. Ágústsson, S. Wiemer, “Imaging high-temperature geothermal reservoirs with ambient seismic noise tomography, a case study of the Hengill geothermal field, SW Iceland”, *Geothermics*, vol. 96, p. 102207, 2021.

A. Obermann, T. Planès, E. Larose, C. Sens-Schönfelder and M. Campillo, "Depth sensitivity of seismic coda waves to velocity perturbations in an elastic heterogeneous medium," *Geophysical Journal International*, vol. 194, p. 372–382, 2013.

16th Oct 23

Dear Dr Sánchez-Pastor,

Your manuscript titled "Monitoring gas pockets in the crust with seismic noise" has now been seen by our reviewers, whose comments appear below. In light of their advice we are delighted to say that we are happy, in principle, to publish a suitably revised version in Communications Earth & Environment under the open access CC BY license (Creative Commons Attribution v4.0 International License).

We therefore invite you to revise your paper one last time to address the remaining concerns of our reviewers. At the same time we ask that you edit your manuscript to comply with our format requirements and to maximise the accessibility and therefore the impact of your work.

Please note that it may still be possible for your paper to be published before the end of 2023, but in order to do this we will need you to address these points as quickly as possible so that we can move forward with your paper.

EDITORIAL REQUESTS:

*****Please take care to match our formatting and policy requirements. We will check revised manuscript and return manuscripts that do not comply. Such requests will lead to delays. *****

SUBMISSION INFORMATION:

OPEN ACCESS:

Communications Earth & Environment is a fully open access journal. Articles are made freely accessible on publication under a [CC BY license](http://creativecommons.org/licenses/by/4.0) (Creative Commons Attribution 4.0 International License). This license allows maximum dissemination and re-use of open access materials and is preferred by many research funding bodies.

For further information about article processing charges, open access funding, and advice and

support from Nature Research, please visit <https://www.nature.com/commsenv/article-processing-charges>

At acceptance, you will be provided with instructions for completing this CC BY license on behalf of all authors. This grants us the necessary permissions to publish your paper. Additionally, you will be asked to declare that all required third party permissions have been obtained, and to provide billing information in order to pay the article-processing charge (APC).

[link redacted]

Best regards,

Carolina Ortiz Guerrero
Associate Editor
Communications Earth & Environment

REVIEWERS' COMMENTS:

Reviewer #1 (Remarks to the Author):

I've read the reply from the authors and the changes to the manuscript and I feel that the authors addressed my remarks properly. The manuscript should be ready for acceptance.

Congratulations on this very interesting work!

Reviewer #2 (Remarks to the Author):

The authors provide additional detailed information to clarify concerns raised by myself and the other reviewer. I am satisfied with the revision.